# Ultraviolet radiation drives mutations in a subset of mucosal melanomas

Piyushkumar A. Mundra [1,11], Nathalie Dhomen[1,11], Manuel Rodrigues [2,3], Lauge Hjorth Mikkelsen [4], Nathalie Cassoux[5], Kelly Brooks[1,6], Sara Valpione[1,7], Jorge S. Reis-Filho [8], Steffen Heegaard[4], Marc-Henri Stern [2,9], Sergio Roman-Roman[10] & Richard Marais [1✉]

Although identified as the key environmental driver of common cutaneous melanoma, the role of ultraviolet radiation (UVR)-induced DNA damage in mucosal melanoma is poorly defined. We analyze 10 mucosal melanomas of conjunctival origin by whole genome sequencing and our data shows a predominance of UVR-associated single base substitution signature 7 (SBS7) in the majority of the samples. Our data shows mucosal melanomas with SBS7 dominance have similar genomic patterns to cutaneous melanomas and therefore this subset should not be excluded from treatments currently used for common cutaneous melanoma.

[1] Molecular Oncology Group, Cancer Research UK Manchester Institute, The University of Manchester, Alderley Park SK10 4TG, UK. [2] Institut Curie, PSL Research University, INSERM U830, DNA Repair and Uveal Melanoma (D.R.U.M.), Equipe labellisée par la Ligue Nationale contre le Cancer, 75248 Paris, France. [3] Institut Curie, PSL Research University, Department of Medical Oncology, 75248 Paris, France. [4] Department of Pathology/Eye Pathology Section, University of Copenhagen, Rigshospitalet 2100 Copenhagen, Denmark. [5] Institut Curie, PSL Research University, Department of Ocular Oncology, 75248 Paris, France. [6] QIMR Berghofer Medical Research Institute, Brisbane, Queensland 4006, Australia. [7] The Christie NHS Foundation Trust, Manchester M20 4GJ, UK. [8] Experimental Pathology Service, Department of Pathology, Memorial Sloan Kettering Cancer Center, New York City, NY, USA. [9] Institut Curie, PSL Research University, Department of Genetics, 75248 Paris, France. [10] Institut Curie, PSL Research University, Translational Research Department, 75248 Paris, France. [11] These authors contributed equally: Piyushkumar A. Mundra, Nathalie Dhomen. ✉email: richard.marais@cruk.manchester.ac.uk

Melanomas are a heterogeneous group of tumors with distinct genomic features that may be broadly classed as epithelium-associated melanomas (includes cutaneous, acral, and mucosal melanomas) or non-epithelium associated melanomas (includes uveal and leptomeningeal melanoma)[1]. Non-epithelium associated melanomas have distinct clinical and genomic features[1,2], but even among epithelium-associated melanomas, the relative frequency and combinations of genomic alterations vary between subtypes. For example, KIT and SF3B1 mutations are more common in mucosal melanomas, whereas BRAF and NRAS mutations are more common in common cutaneous melanomas[1].

Ultraviolet radiation (UVR)-induced DNA damage is a clinically relevant factor that distinguishes the different melanoma subtypes[3]. It is clearly linked to the development of common cutaneous melanomas, but its contribution to the rarer subtypes is largely assumed to be negligible, because they tend to arise in sun-protected tissues. Mucosal melanoma (1.4% of melanomas) is an example of such a rare melanoma subtype, which arises in the mucosa of the eyes, mouth, nose, and gastrointestinal and genitourinary tracts. It presents distinct biological and clinical features, responds poorly to treatment, and is characterized by distinct genomic traits, with high numbers of chromosomal structural changes, low mutation burden, and specific patterns of driver oncogenes[4–8]. This is thought to be because mucosal melanomas arise in distinct microenvironments and are not driven by UVR[1]. However, two recent studies[7,9] reported that 9% (6/67, 6/65 respectively) of mucosal melanomas present >50% of COSMIC single base substitution signature 7 (SBS7), a mutation signature associated with UVR[10]. It was also recently reported that uveal melanoma, another rare melanoma subtype not generally associated with UVR-exposure, can present SBS7-predominance if it arises on the iris[11]. We hypothesized that UVR drives melanomagenesis independent of tissue microenvironment, so to test this we performed whole-genome sequencing (WGS) on mucosal melanomas from the conjunctiva, because this tissue is largely UVR exposed[12].

In this study, we present a comparison of the genomic landscape of these UVR-exposed mucosal melanomas to other primary mucosal melanomas and to primary cutaneous melanomas, to provide better understanding of the oncogenes and mutational processes that drive this particular melanoma subtype.

## Results

### UVR-driven DNA damage is predominant in mucosal melanomas of conjunctival origin. We performed WGS on 10 fresh frozen primary conjunctival melanomas (median patient age 66 years, range 38–84; six females, four males; Supplementary Table 1) and compared our results to published WGS from eight mucosal melanomas originating on sun-protected sites (nasal, genitourinary, rectal; median patient age 63 years, range 46–84; six females, two males; Supplementary Table 2)[6]. Mutational signature analysis on our WGS data revealed a predominance of COSMIC SBS7v2 in nine of the conjunctival melanomas, whereas one conjunctival melanoma and the other eight mucosal melanomas were dominated by the SBS1v2 (age-related), SBS5v2 (ubiquitous), and SBS3v2 (BRCA1, BRCA2, and PALB2-associated) (Fig. 1a, b). To facilitate direct comparison with common cutaneous melanoma, we used our pipeline to analyze published WGS from 54 primary common cutaneous melanomas[6] and observed SBS7v2 predominance in 51 of these samples (Fig. 1a, b). Compared to their non-SBS7v2 counterparts, the SBS7v2 mucosal melanomas presented higher proportions of C-to-T transitions at dipyrimidines (mean 84.7% versus 29.0%; P < 0.0001; Fig. 1c, d) and higher numbers of single nucleotide

variants (SNVs) (median 100,098 [range 42,649–274,061] vs. 10,391 [range 8426–20,538]; P < 0.0001; Fig. 1e, f). Similarly, compared to their non-SBS7v2 counterparts, the SBS7v2 cutaneous melanomas presented higher proportions of C-to-T transitions at dipyrimidines (median 82.34% versus 36.09%; P < 0.0001; Fig. 1c, d) and higher numbers of SNV (median 119,058 [range 20,021–938,462] vs. 11716 [range 9838–12,607]; P < 0.0001; Fig. 1e, f). Notably, the SBS7v2 mucosal and common cutaneous melanomas presented similar proportions of SNV and C-to-T transitions at dipyrimidines (Fig. 1b, d, f). Equally, the non-SBS7v2 mucosal and common cutaneous melanomas presented similar proportions of SNVs, and similar proportions of C-to-T transitions at dipyrimidines (Fig. 1b, d, f) and other nucleotide transitions/transversions (Supplementary Fig. 1a–e). Thus, nine conjunctival mucosal melanomas exhibited features of UVR exposure, whereas one conjunctival and the other eight mucosal melanomas did not present these features.

We validated our findings in an independent cohort of 65 published mucosal melanoma whole genomes[9]. Here we identified eight samples (12%) with SBS7v2 predominance (Supplementary Fig. 2a), which also presented higher SNV loads (Supplementary Fig. 2b), higher proportions of C-to-T transitions at dipyrimidines, and lower proportions of other transitions/transversions than their non-SBS7v2 counterparts (Supplementary Fig. 2c–h).

### Structural variants distinguish mucosal melanomas from cutaneous melanomas and are independent of UVR mutation signature status. Previous studies have reported increased numbers of structural variants and indels in mucosal melanomas compared to common cutaneous melanomas[6]. We investigated whether UVR-induced DNA damage influenced the extent or pattern of structural variation in mucosal melanomas. Consistent with previous studies[6,7], the mucosal melanomas presented more structural variants and indels than common cutaneous melanomas (Fig. 2a). Critically, we observed no significant differences between the SBS7v2-dominant and the non-SBS7v2 cohorts (Fig. 2a and Supplementary Fig. 3a), and similarly no significant differences in copy number variations (Fig. 2b). This was recapitulated in the validation cohort, where we again observed no significant difference in chromosomal structural variants or number of indels between the SBS7v2-dominant and non-SBS7v2 mucosal melanomas (Supplementary Fig. 3b, c).

### SBS7 dominance in mucosal melanoma is a better indicator of UVR-exposure than tumor site. Large areas of the conjunctiva are highly sun-exposed, and this is reflected in the SBS7v2 dominance in nine of ten genomes from our primary conjunctival melanomas. MuM12, MuM13, MuM16, and MuM17 were localized to the limbus and MuM14 to the upper part of the bulbar conjunctiva, areas that are frequently sun-exposed (Supplementary Table 1). Note, however, that SBS7v2 also dominated the genomes of MuM10, MuM11, and MuM18, which were from the tarsal conjunctiva which is considered to be more sun-protected, and SBS7v2 dominated the genome of MuM15, which was a large lesion spanning the sun-protected fornix and the sun-exposed caruncle (Supplementary Table 1). Note also that MuM1, which presented the lowest mutation burden and was the only primary conjunctival melanoma that did not exhibit SBS7v2 dominance, arose from the fornix, considered to be the most sun-protected part of the conjunctiva. Similarly, the conjunctival melanoma with the next lowest mutation burden and SBS7v2 contribution was MuM10, another forniceal tumor.

In our validation cohort, the SBS7v2-dominant mucosal melanomas were from potentially sun-exposed sites, including

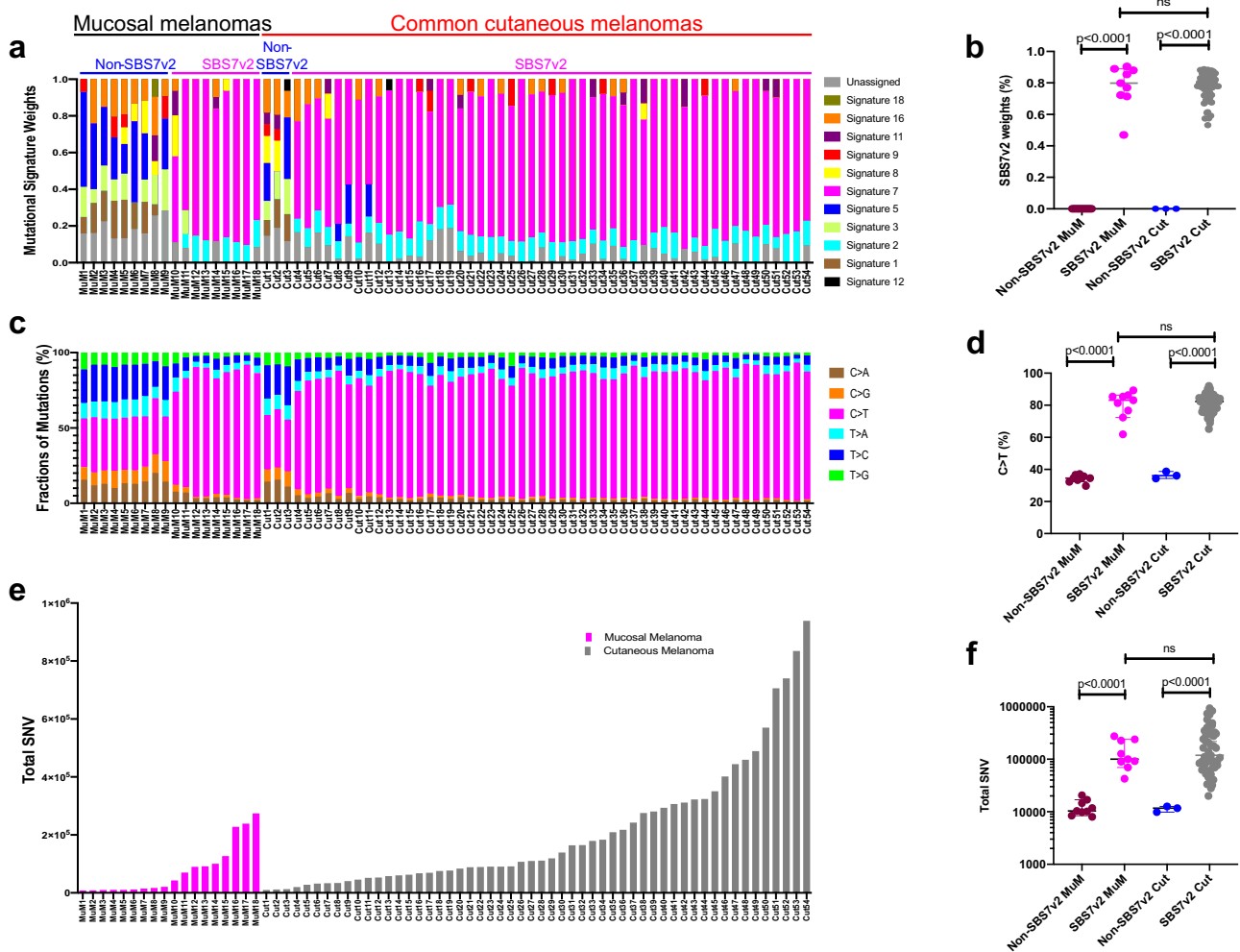

**Fig. 1 Mutation spectra in mucosal and common cutaneous melanomas. a** Mutation signatures weighted by relative contribution to spectrum of mutations in individual tumor genomes. Indices above indicate mucosal (black, $n = 18$) or common cutaneous (red, $n = 54$) melanomas with subdivision into non-SBS7v2 (blue, $n = 9$, $n = 3$, respectively) and SBS7v2 (magenta, $n = 9$, $n = 51$ respectively) genomes. Columns represent individual tumors. **b** Relative contribution of SBS7v2 to mutation spectra (SBS7v2 Weights) for individual mucosal (MuM) or cutaneous (Cut) melanomas with SBS7v2 (magenta $n = 9$, gray $n = 51$) or non-SBSv2 (maroon $n = 9$, blue $n = 3$) genomes. **c** Proportion of six nucleotide transitions/transversions for individual tumor genomes (individual columns). **d** Proportions of C > T nucleotide transitions in individual mucosal (MuM) or cutaneous (Cut) melanomas with SBS7v2-dominant (magenta $n = 9$, gray $n = 51$) or non-SBSv2 (maroon $n = 9$, blue $n = 3$) genomes. **e** Total SNVs in individual mucosal (MuM, magenta, $n = 18$) or cutaneous (Cut, gray, $n = 54$) melanomas. Columns represent individual tumors. **f** Total SNVs in mucosal (MuM) and cutaneous (Cut) melanomas with SBS7v2-predominant (magenta $n = 9$, gray $n = 54$) or non-SBSv2 (maroon $n = 9$, blue $n = 3$) genomes. Panels **b**, **d**, **f** show median and 95% confidence intervals, dots denote individual tumors, $P$-values determined by two-tailed Mann–Whitney $U$, ns = not significant: 0.4140 in **b**, 0.8387 in **d**, 0.6838 in **f**, respectively.

the lips (3/5), gingiva (2/28), nasal cavity (1/2), multi-sites (lip and gingiva) (1/2), and oropharynx (1/1)[9]. Thus, in these samples also, SBS7v2-dominant mucosal melanomas were largely from potentially sun-exposed sites, but it should be noted that the SBS7v2-domimant mucosal melanomas were still in the minority. Thus, although the precise location of these melanomas is not known, our analysis suggests that the SBS7v2 dominance is a more specific marker of UVR-driven processes than tumor site and henceforth we refer to these tumors as UVR-exposed mucosal melanomas.

**UVR-exposed mucosal and cutaneous melanomas present similar driver oncogene mutations**. We next investigated mutations in common melanoma-associated oncogenes. *BRAF* mutations are generally associated with common cutaneous melanoma, but only weakly associated with mucosal melanoma[1].

We observed that six of the nine UVR mucosal melanomas carried *BRAF* mutations, whereas only one of the nine non-UVR mucosal melanomas had a *BRAF* mutation (Fig. 2c). Notably, eight of the nine UVR-exposed mucosal melanomas and all 51 UVR-exposed cutaneous melanomas carried mutations in one to eleven known melanoma genes, with the remaining mucosal melanoma (MuM10) carrying a frame-shift mutation in the melanocyte gene *TYRP1* (Fig. 2c and Supplementary Data 1). Thus, *FGFR2/4, ERBB4, NF1, CDKN2A, NFKBIE, SALL4, TERT, GRIN2A,* and *TP53* mutations were restricted to UVR-exposed melanomas (Fig. 2c). Conversely, the non-UVR-exposed mucosal and cutaneous melanomas carried mutations in only two (1 sample), one (5 samples), or none (6 samples) of the melanoma genes (Fig. 2c). Additionally, 31 (52%) UVR-exposed melanomas had *TERT* and/or *TERT* promoter mutations, but only one (8%) non-UVR-exposed melanoma had a mutation in this gene (Fig. 2c, d). These data show remarkable enrichment for the same

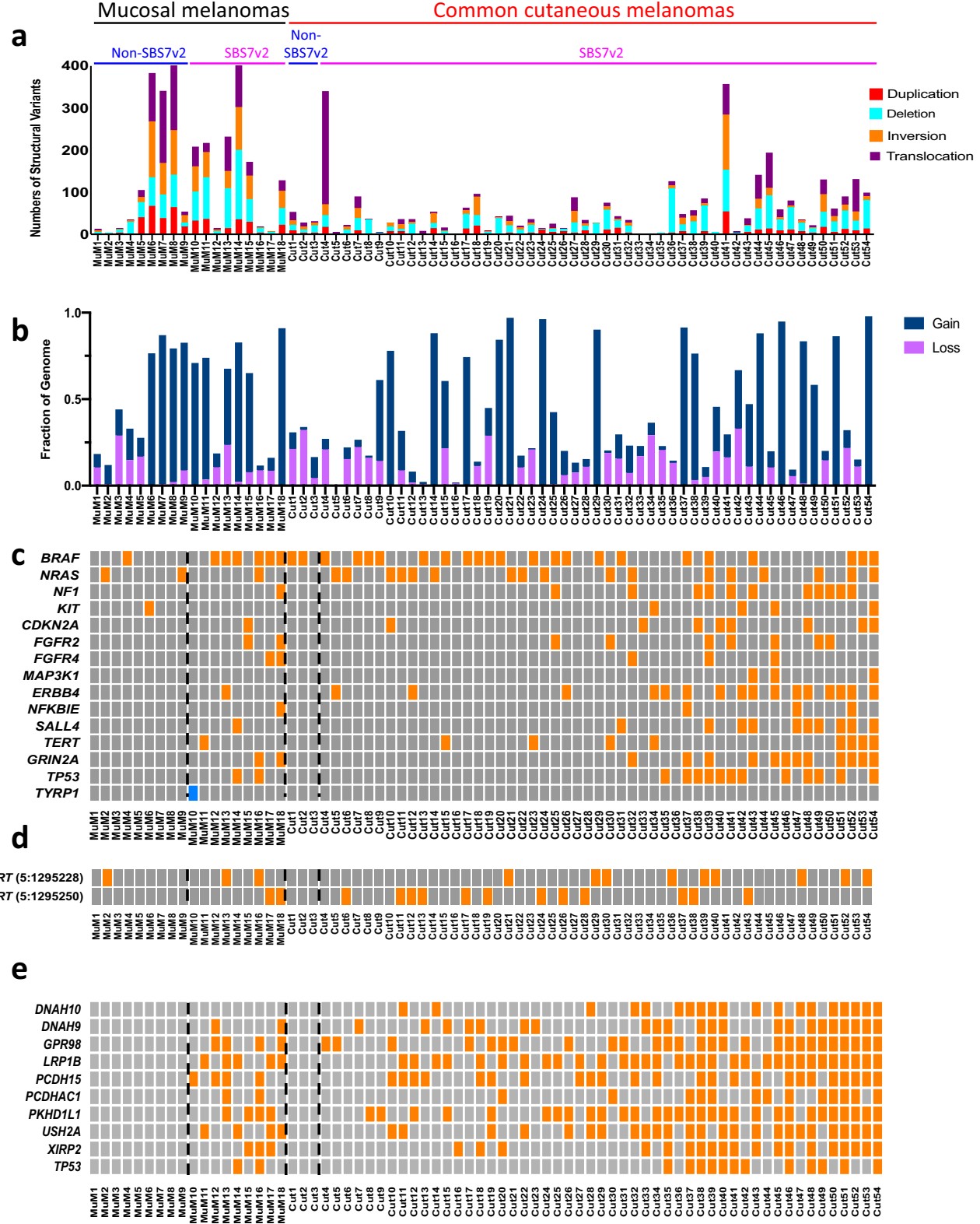

**Fig. 2 Structural alterations and gene mutations in mucosal and common cutaneous melanomas.** Numbers and types of structural variants in individual melanoma genomes. Indices above indicate mucosal (black, $n = 18$) and common cutaneous (red, $n = 54$) melanomas with subdivision into non-SBS7v2 (blue; $n = 9$, $n = 3$, respectively) and SBS7v2-dominant (magenta; $n = 9$, $n = 51$, respectively) genomes. Columns represent individual tumors. **b** Relative amounts of chromosomal gains and losses (fraction of genome) in individual genomes. Columns represent individual tumors. **c**, **d** Missense (orange) or frameshift (blue) mutations in known melanoma oncogenes (**c**), or *TERT* promoter regions (**d**) of individual tumors. **e** Missense mutations (orange) in the indicated genes for individual tumors (columns). Dotted lines denote segregation into non-SBS7v2 and SBS7v2 dominant genomes.

driver oncogene mutations in UVR-exposed cutaneous melanoma and UVR-exposed mucosal melanoma.

**Ten-gene panel as surrogate to UVR signature in mucosal melanomas.** We previously reported that mutations in a ten-gene panel (*LRP1B, GPR98, XIRP2, PKHD1L1, USH2A, DNAH9, PCDH15, DNAH10, TP53, PCDHAC1*) were a surrogate of UVR exposure[3]. We therefore investigated if this panel could segregate UVR-exposed from non-UVR exposed mucosal melanomas. Remarkably, this panel correctly segregated 71/72 (97%) of the UVR-exposed mucosal and cutaneous melanomas (Fig. 2e), including all nine UVR mucosal melanomas, eight of which carried mutations in two or more of these genes (Fig. 2e). This panel therefore provides a targeted approach that allows rapid screening for UVR-exposed mucosal melanomas.

## Discussion

Although our cohort was small due to the challenges inherent in obtaining samples, we present WGS for ten conjunctival melanomas, and our results extend previous studies by showing that conjunctival mucosal melanomas have similar genomes to common cutaneous melanoma[12–14]. We also show that like common cutaneous melanoma[3], mucosal melanomas present two broad groups, one with SBS7v2 predominance that appears to be UVR-driven, and one that is not UVR-driven, but curiously, both groups present the large structural alterations more commonly observed in mucosal melanoma. Our analysis revealed particularly striking similarities between UVR-exposed mucosal melanomas and UVR-driven common cutaneous melanomas, as both presented high mutation burdens and abundant mutations that activate the BRAF–ERK pathway.

Notably, *BRAF* mutations are rare in mucosal melanoma compared to common cutaneous melanoma[15,16], so *BRAF* mutation testing is not recommended or reimbursed in some jurisdictions. However, we show that UVR-driven mucosal melanomas harbor high frequency *BRAF* mutations, so could benefit from BRAF and MEK targeted therapies. We note that the first published case of a patient with a *BRAF*-mutated metastatic conjunctival melanoma treated with vemurafenib was reported 7 years after the first clinical results of this drug[17], and 5 years after FDA approval for cutaneous melanoma patients[18]. Moreover, although *CKIT* mutations are present in about 15% of mucosal melanomas[1] response rates to KIT inhibitors range from only 5 to 26%, and no KIT drugs are approved for use in these patients.

Mucosal melanoma patients are unfortunately also generally excluded from immunotherapy trials[19–21] and because of this exclusion, immunotherapies are not approved for mucosal melanoma in the adjuvant setting. Note, however, that response rates to immune checkpoint inhibitors in advanced mucosal melanomas are at least half of that seen in non-glabrous skin melanomas[22,23], and it was recently reported that four of five conjunctival melanoma patients responded to immunotherapy[24]. The clear similarities between UVR-driven cutaneous melanoma and UVR-associated mucosal melanoma suggest that mucosal melanoma patients with SBS7v2 predominance may benefit from BRAF/MEK inhibitor combinations and from immunotherapies in both the advanced and adjuvant settings.

Despite the similarities in mutation burden and oncogene pathway activation, we note that chromosomal structural variations did not distinguish UVR from non-UVR mucosal melanomas, but did distinguish mucosal from common cutaneous melanoma. This suggests that UVR imposes additional processes over the microenvironment-specific mutational processes that otherwise drive the different melanoma subtypes. Together, these data show us that mucosal melanomas do not belong to a homogeneous group of diseases and suggest that tumors arising from mucosal melanocytes are subject to a common tumorigenic process resulting in the accumulation of structural genome variations, to which are added UVR-driven processes in a subset of cases. This aligns with recent reports that *SF3B1* R625 mutations are recurrently present in vulvo-vaginal and anorectal melanomas but not in other mucosal locations[25,26]. Our study therefore contributes to the definition of biologically and clinically relevant subsets of mucosal melanomas, providing better insight into their biology and opening avenues for precision medicine.

The UVR-driven mucosal melanomas tend to arise on sun-exposed sites such as the conjunctiva and lips, but sun exposure per se does not identify this subset of mucosal melanomas, as highlighted by the presence of non-SBS7v2 tumors at potentially sun-exposed sites and SBS7v2 tumors at more sun-protected sites. Some of our SBS7v2 cases came from gingiva, oropharynx, or nasal cavity. Whilst imprecision in site reporting may play a role in this, one possibility is that mucosal cells in sun-exposed sites may accumulate UVR-induced mutations and expand into large clones of mutant cells, extending into sun-protected areas where such cells could further develop into a melanoma. Clonal expansions of this nature have been reported in the skin[27] and in Barrett's esophagus[28]. Conversely, the presence of non-SBS7v2 tumors at potentially sun-exposed sites is consistent with our previously report that in a UVR/BRAF^V600E-driven mouse melanoma model, a small number of tumors developed without evidence of UVR-associated DNA damage[3]. As outlined above, distinguishing UVR and non-UVR melanomas in the mucosal and other settings could have important implications for clinical care. Our data shows that this cannot be determined by the site of the primary tumor, but we propose that our 10-gene panel could provide rapid testing for the UVR signature without the cost and complexity of WGS. Together with the knowledge that the UVR-associated mucosal melanomas could benefit from treatments currently used in common cutaneous melanoma, our findings highlight an approach to delivering precision medicine to this patient group for whom current treatment options are limited.

## Methods

**Sample collection and ethics.** Conjunctival melanoma samples MuM1 and MuM10–18 comprise two cohorts, one from Institut Curie, Paris (MuM10, MuM12, MuM15–18) and one from the department of Pathology/Eye Pathology Section, University of Copenhagen, Rigshospitalet (MuM1, MuM11, MuM13, MuM14). The studies were approved by the Internal Review Board of Institut Curie (2014) and the Danish National Ethics Committee (j. no. 1700673), respectively. The samples were collected during surgery and frozen immediately. Paired blood samples were also collected and stored for sequencing. Patients provided written informed consent to perform germline and somatic genetic analyses for WGS on archived frozen samples.

**DNA extraction and WGS.** For Institut Curie cohort, DNA was extracted by the *Centre de Ressources Biologiques* (Institut Curie tumor biobank) from frozen samples using phenol (Invitrogen), then subsequently purified on Zymo-Spin IC (Zymo Research). DNA was extracted from paired whole blood samples using the QuickGene DNA whole-blood kit with QuickGene-610L equipment (Fujifilm, Japan). DNA concentrations were quantified by Qubit (Thermo Fisher Scientific). Integrities were assessed by a BioAnalyzer 2100 device (Agilent Technologies, Santa Clara, CA, USA). For Rigshospitalet cohort, DNA/RNA was extracted from frozen tumor samples using Norgen (Biotek Corp.) kit and from blood samples using QIAamp DNA Blood and Tissue kit (Qiagen, Manchester, UK) as per manufacturer's instructions. Concentrations were quantified by Qubit (Thermo Fisher Scientific).

**Sequencing and processing.** WGS of the tumor-normal pairs from patients included in the Institut Curie cohort (MuM10, MuM12, MuM15-18) was performed in the New York Genome Center (NYGC). DNA libraries were prepared using TruSeq PCR-free approach following the protocols implemented in the NYGC. WGS of the tumor-normal pairs of the Rigshospitalet cohort (MuM1, MuM11, MuM13, MuM14) was performed by Edinburgh Genomics (The Roslin Institute, University of Edinburgh) using TruSeq Nano library preparation method.

Sequencing was performed on Illumina HiSeqX machine for both the cohorts with aimed coverage of 30× and 60× for normal (blood) and tumor samples, respectively.

WGS BAM files of other primary mucosal melanomas (MuM2–MuM9) and primary cutaneous melanomas[6] (Cut1-Cut54) were downloaded from EGA using accession ID EGAS00001001552 using ASPERA (v3.5.4).

**Bioinformatics analysis**. FASTQ files were extracted for BAM files for MuM2–MuM9 and cutaneous melanoma samples using samtools[29] (v1.3.1). The 2 × 150 read pairs were mapped to the reference genome GRCh37 (v75) using BWA-mem[30] (v0.7.7) tool. This was followed by duplicate removal using PICARD (v1.96) and INDELs realignment and recalibration of base qualities using GATK[31] (v3.6).

The final BAM files from both tumor and corresponding blood sample were used for subsequent somatic variant calling using MUTECT[32] (v1.1.7) with default parameters. Small insertions and deletions were determined using Strelka[33] (v1.0.4). Only "Passed" calls were considered. Variant effect predictor[34] (Ensembl version 73) was used to annotate the mutations. Known variants present in dbSNP were excluded. Structural variants were determined using DELLY[35] (v0.8.1) with default parameters.

Mutational signatures were determined by fitting somatic SNVs with trinucleotide context to the 30 COSMIC mutational signatures using deconstructSigs (v1.8.0)[36] package using default parameters. Signatures with contribution weights less than 6% were excluded.

Copy number alterations were determined using Sequenza[37] (v2.1.9999b0) package with parameters (mufreq.treshold = 0.05, min.reads = 10, min.fw.freq = −0.1). One cutaneous sample with missing gender information in clinical data was excluded from the analysis. Fraction of genomic alteration for each sample was calculated using an in-house script.

**Reporting summary**. Further information on research design is available in the Nature Research Reporting Summary linked to this article.

## Data availability
The whole-genome sequencing data generated in this study from conjunctival melanoma samples have been deposited in the EGA database under accession code EGAS00001004697. The data is available under restricted access, which can be obtained by contacting Prof. Richard Marais. The whole-genome sequencing data accessed for use in this study corresponding to cutaneous melanoma and other mucosal melanoma samples are available from the EGA database under accession code EGAS00001001552. The remaining data are available within the Article, Supplementary Information, or available from the author upon request.

## Code availability
In-house codes used to compute fraction of genome altered are available at: https://github.com/mpiyush21/MucosalNatureComms.

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

## Acknowledgements
R.M. is supported by Wellcome Trust (100282/Z/12/Z), European Research Council (ERC-ADG-2014 671262), Cancer Research UK (A27412 and A22902). M.R. and M.-H.S. are funded by Institut Curie, the Ligue Nationale Contre le Cancer (Labellisation), and Site de Recherche Intégrée sur le Cancer (SiRIC) de l'Institut Curie. M.-H.S. is also funded by Institut National de la Santé et de la Recherche Médicale. R.M., M.-H.S., and S.R. are supported by the European Commission under the Horizon 2020 program (U.M. Cure; project number: 667787). L.H.M. was funded by a non-restricted Candys Foundation grant. J.S.R.-F. is funded in part the Breast Cancer Research Foundation. S.V.

is supported by a Harry J Lloyd Charitable Trust Career Development Award. WGS costs were partly supported by the Manchester NIHR Biomedical Research Centre.

## Author contributions

Conceptualization, P.A.M., M.R., L.H.M., N.D., and R.M.; Methodology, K.B., S.V., N.C., J.S.R.-F.; Formal analysis, P.A.M., M.R., L.H.M., and N.D.; Resources, M.R., L.H.M., N.C., S.H., M.-H.S., S.R., J.S.R.-F.; Writing, P.A.M., N.D., and R.M.; Supervision, S.H., M.-H.S., S.R., and R.M.

## Competing interests

R.M. consultants for Pfizer, and as a former Institute of Cancer Research (London) employee, he may benefit from commercialized programs. J.S.R.F. consultants for Goldman Sachs, REPARE Therapeutics and Paige.AI, and serves on the scientific advisory boards of Volition Rx and Paige.AI, and ad hoc on the scientific advisory boards of Ventana Medical Systems, Roche Tissue Diagnostics, Genentech, Novartis, and InVicro. All other authors declare no competing interests.
