## [Peer Review File · Nature Communications]

REVIEWER COMMENTS

Reviewer #1 (Remarks to the Author): Expert in melanoma genomics

The investigators describe the results of WGS of 10 primary conjunctival melanoma and compared the data to published WGS. The major finding is that 9 out of 10 conjunctival melanoma display predominantly UVR signature mutations (COSMIC SBS7v2). The investigators agree that these are not surprising results as conjunctival melanomas reside in sun exposed site, and that published data in which 67 mucosal melanoma were analyzed showed UVR-related mutations in the 2 conjunctival melanomas and as well as in other facial areas (Nat Commun, 2019, PMC6639323).

The investigators also suggest that these data support the notion that “patients with UVR-driven mucosal melanomas could benefit from the therapies given to patients with common cutaneous melanoma”. Unfortunately, mutation rates and kind of mutations are not a factor in choosing treatment of all melanoma patients. Currently, in addition to surgery, the established systemic treatments are immunotherapy or targeted therapy, such as BRAF and KIT to ALL melanoma patients.

Reviewer #2 (Remarks to the Author): Expert in UV radiation response

Summary:

Sunlight is the main cause of cutaneous melanomas, but mucosal melanomas have been considered unrelated to sunlight. The authors noticed that previous DNA sequencing studies of mucosal melanomas indicated that a fraction of them contained UV signature mutations, so the present study examined mucosal melanomas at specific body sites in greater depth. The studies included both eye conjunctiva samples collected by the authors and re-analysis of publicly archived datasets that included other body sites.

The authors found that mucosal melanomas on the eye conjunctiva have a preponderance of UV-signature mutations. Subsequent examination of literature datasets indicated the presence of UV-signature mutations in mucosal melanomas of the lips, gingiva, hard palate, and nasal cavities. Combining the samples from these UV-signature tumors revealed that the total mutation burden was 10-fold higher than in mucosal melanomas from sun-shielded body sites and the genes mutated resembled those mutated in cutaneous melanomas. These properties were not the case in mucosal melanomas from vulva or rectum and they resemble properties of cutaneous melanomas. The known higher frequency of large-scale chromosomal alterations in mucosal melanomas did not differ between these mucosal melanomas and those from sites that are clearly sun-shielded. The authors conclude that sunlight UV can drive melanoma formation in mucosal melanomas where it reaches these cells and suggest that therapies used for cutaneous melanomas might be efficacious in this class of mucosal melanomas.

Critique:

The studies are carefully executed and analyzed, the result is somewhat surprising, and the suggested approach to therapy is plausible. This is a significant paper.

However, the identification of just what subset of mucosal melanomas is sun-driven is unclear in both the explanation and the reasoning. This can be fixed with some re-writing.

1. Xeroderma pigmentosum patients, defective in repair of UV photoproducts, have 1000x elevated rates of melanoma and non-melanoma skin cancers, and 100,000x elevated squamous cell carcinoma on the tip of the tongue, but they don't have elevated cancer frequencies in the gingiva, oral cavity, or nasal cavity (eg DiGiovanna & Kraemer, JID 132;785, 2012 plus some personal communications). So those sites are apparently not actually sun-exposed. Thus it is not clear that sunlight-exposure of the oral or nasal cavity led to these tumors.

2. line 91. In the dataset from the Zhou paper, 8 of 65 mucosal melanomas were found to have UV signatures and other UV-like properties. These 8 were "from potentially sun-exposed sites, including the lips, gingiva, nasal cavity and oropharynx". On its face, this is plausible. But the other 57 mucosal tumors were not from vulva and colon. Not stated in the paper is the fact that 55 of the other 57 mucosal tumors were also from the lips, gingiva, nasal cavity and oropharynx. 5 total were from the lip, so 58 were from gingiva, nasal cavity, and oropharynx. (I can't tell from the presentation whether all 5 lip melanomas had UV signature mutations.)

Therefore the situation is not simply that sunlight entering the oral cavity explains mucosal melanomas.

3. Two potential explanations are:

a) The UV-related oral/nasal mucosal melanomas are metastases from a sun-exposed site such as the lip.

b) Something in the Chinese diet, or smoking or a habit like chewing betel nuts, triggered chemically-induced UV-like DNA photoproducts in these patients, analogous to the later steps of a mechanism recently reported (Science 347:842, 2015). This is more hypothetical, as it would require a previously unknown trigger and it leaves unexplained the oral/nasal melanomas that didn't have UV signatures.

4. At a minimum, the paper needs to state that only a small fraction of the mucosal melanomas of gingiva, nasal cavity, and oropharynx were sun-related and that this is a bit of a mystery because the xeroderma pigmentosum data suggest that these sites are not directly exposed to sunlight.

5. Whatever the biological explanation, it does remain clinically useful to identify those mucosal melanomas that have sunlight-like mutations, using the small gene panel that the authors describe, because these appear to also have mutated genes for which there are therapeutic approaches.

I also have a two suggestions for conveying the message more clearly.

Title:

The reader will assume that the term "subtype" means a histological subtype, but what is meant is "a subset of mucosal melanomas". For the reasons outlined above, one can't even say "on sun-exposed body sites", although that would be true for conjunctiva and lip.

Abstract and Conclusion:

"... could benefit from treatments currently used for cutaneous melanoma".

My impression is that melanoma treatments are not chosen based on a UV origin of their mutations.

The authors need to clarify whether they are thinking that:

a) the same genes are hit as in cutaneous melanoma and so targeted therapies might work in these patients

or

b) the high mutation burden suggests that immunotherapies might be effective in these patients.

Reviewer #3 (Remarks to the Author): Expert in UV response and melanoma genomics

Mucosal melanomas are conventionally considered a separate "species" of melanomas that are distinct from the cutaneous melanomas. In this manuscript, the authors have presented whole genome sequencing analysis of conjunctival mucosal melanomas and shown that majority of these melanomas harbor mutational signatures that suggest that these melanomas are driven by ultraviolet radiation (UVR), akin to their cutaneous counterparts, but unlike the mucosal melanomas from sun-protected areas. The authors conclude that UVR induces melanomagenesis independently of the tissue microenvironment. The study design, implementation, data acquisition and analyses, and interpretations of the data are of high quality.

However, this study is entirely descriptive with a predictable outcome, i.e. that UVR-exposed mucosal melanomas harbor UVR mutational signatures. The paper lacks novelty, innovation, and palpable biological insights.

We thank the reviewers for their insightful comments, which have helped us to improve the quality of the manuscript. Below we provide responses to each comment.

Reviewer #1

Comment 1: *The investigators describe the results of WGS of 10 primary conjunctival melanoma and compared the data to published WGS. The major finding is that 9 out of 10 conjunctival melanoma display predominantly UVR signature mutations (COSMIC SBS7v2). The investigators agree that these are not surprising results as conjunctival melanomas reside in sun exposed site, and that published data in which 67 mucosal melanoma were analyzed showed UVR-related mutations in the 2 conjunctival melanomas and as well as in other facial areas (Nat Commun, 2019, PMC6639323).*

Response: We thank the referee for their constructive feedback. Notably, we also observe a lack of UV associated mutational signature in one conjunctival melanoma and several mucosal melanomas occurring in facial areas. Hence, as Reviewer 2 noted, mucosal melanomas at potentially sun exposed sites don't necessarily exhibit UVR-associated DNA damage. Thus, whilst we posited that UVR-associated DNA damage would occur in conjunctival melanomas, we also clarify below and in our revised manuscript that the location of a melanoma (whether cutaneous or mucosal) at potentially sun-exposed sites does not, *per se*, correlate with UVR-associated DNA damage as determined by the predominance of single base substitution signature 7 (SBS7). Currently, the SBS7 dominant and non-SBS7 groups in both cutaneous or mucosal melanoma are only distinguishable by genomic analysis. However, we recently clarified that in cutaneous melanoma SBS7 defines distinct groups with different clinical outcomes (Trucco et al, 2018, *Nature Medicine*, 2019 Feb;25(2):221-224) and here we show that mucosal melanomas also fall into these distinct categories. Our findings therefore extend observations from cutaneous melanoma and as we note below, this could have important clinical implications.

Comment 2: *The investigators also suggest that these data support the notion that “patients with UVR-driven mucosal melanomas could benefit from the therapies given to patients with common cutaneous melanoma”. Unfortunately, mutation rates and kind of mutations are not a factor in choosing treatment of all melanoma patients. Currently, in addition to surgery, the established systemic treatments are immunotherapy or targeted therapy, such as BRAF and KIT to ALL melanoma patients.*

Response: We thank the referee for the helpful comment, which prompted us to further clarify the clinical unmet need we address in our study. We have expanded the main text to include an *Introduction* and *Discussion* section to clarify some of the points raised by the referees. This includes the following paragraphs in the *Discussion* (page 8, line 200):

“Notably, *BRAF* mutations are rare in mucosal melanoma compared to common cutaneous melanoma^{15,16}, so *BRAF* mutation testing is not recommended or reimbursed in some jurisdictions. However, we show that UVR-driven mucosal melanomas harbor high frequency *BRAF* mutations, so could benefit from *BRAF* and *MEK* targeted therapies. We note that the first published case of a patient with a *BRAF*-mutated metastatic conjunctival melanoma treated with vemurafenib was published seven years after the first clinical results of this drug¹⁷, five years after FDA approval for cutaneous melanoma patients¹⁸. Moreover, although *CKIT* mutations are present in about 15% of mucosal melanomas¹ response rates to *KIT* inhibitors range from only 5 to 26%, and no *KIT* drugs are approved for use in these patients.

Mucosal melanoma patients are unfortunately also generally excluded from immunotherapy trials¹⁹⁻²¹ and because of this exclusion, immunotherapies are not approved for mucosal melanoma in the adjuvant setting. Note however that response rates to immune checkpoint inhibitors in advanced mucosal melanomas are about half of that seen in to non-glabrous skin melanomas^{22,23}, and it was recently reported that four of five conjunctival melanoma patients responded to immunotherapy²⁴. The clear similarities between UVR-driven cutaneous melanoma and UVR-associated mucosal melanoma suggests that mucosal melanoma patients with SBS7v2 predominance may benefit from *BRAF/MEK* inhibitor combinations and from immunotherapies in both the advanced and adjuvant settings.”

Reviewer #2

Summary:

Sunlight is the main cause of cutaneous melanomas, but mucosal melanomas have been considered unrelated to sunlight. The authors noticed that previous DNA sequencing studies of mucosal melanomas indicated that a fraction of them contained UV signature mutations, so the present study examined mucosal melanomas at specific body sites in greater depth. The studies included both eye conjunctiva samples collected by the authors and re-analysis of publicly archived datasets that included other body sites.

The authors found that mucosal melanomas on the eye conjunctiva have a preponderance of UV-signature mutations. Subsequent examination of literature datasets indicated the presence of UV-signature mutations in mucosal melanomas of the lips, gingiva, hard palate, and nasal cavities. Combining the samples from these UV-signature tumors revealed that the total mutation burden was 10-fold higher than in mucosal melanomas from sun-shielded body sites and the genes mutated resembled those mutated in cutaneous melanomas. These properties were not the case in mucosal melanomas from vulva or rectum and they resemble properties of cutaneous melanomas. The known higher frequency of large-scale chromosomal alterations in mucosal melanomas did not differ between these mucosal melanomas and those from sites that are clearly sun-shielded. The authors conclude that sunlight UV can drive melanoma formation in mucosal melanomas where it reaches these cells and suggest that therapies used for cutaneous melanomas might be efficacious in this class of mucosal melanomas.

Critique:

The studies are carefully executed and analyzed, the result is somewhat surprising, and the suggested approach to therapy is plausible. This is a significant paper.

However, the identification of just what subset of mucosal melanomas is sun-driven is unclear in both the explanation and the reasoning. This can be fixed with some re-writing.

Comment 1: *Xeroderma pigmentosum patients, defective in repair of UV photoproducts, have 1000x elevated rates of melanoma and non-melanoma skin cancers, and 100,000x elevated squamous cell carcinoma on the tip of the tongue, but they don't have elevated cancer frequencies in the gingiva, oral cavity, or nasal cavity (eg DiGiovanna & Kraemer, JID 132:785, 2012 plus some personal communications). So those sites are apparently not actually sun-exposed. Thus it is not clear that sunlight-exposure of the oral or nasal cavity led to these tumors.*

Response: We thank the reviewer for drawing attention to our wording, which was not sufficiently clear in discussing the potential for sun exposure in mucosal melanomas. We did not intend to propose that it is possible to segregate mucosal melanomas as sun-exposed or non-sun exposed based on location. To address this important point, we have added a brief analysis of the proportion of dominant SBS7 VS non-SBS7 tumours in sun-exposed areas in the Results, under the heading: "SBS7 dominance in mucosal melanoma is a better indicator

of UVR-exposure than tumor site" (page: 5, line 136). Additionally, we have included further discussion of this point in the final paragraph of the *Discussion* (starting page: 9 line 234).

Comment 2,3,4 (addressed together):

2. line 91. *In the dataset from the Zhou paper, 8 of 65 mucosal melanomas were found to have UV signatures and other UV-like properties. These 8 were "from potentially sun-exposed sites, including the lips, gingiva, nasal cavity and oropharynx". On its face, this is plausible. But the other 57 mucosal tumors were not from vulva and colon. Not stated in the paper is the fact that 55 of the other 57 mucosal tumors were also from the lips, gingiva, nasal cavity and oropharynx. 5 total were from the lip, so 58 were from gingiva, nasal cavity, and oropharynx. (I can't tell from the presentation whether all 5 lip melanomas had UV signature mutations.)*

Therefore the situation is not simply that sunlight entering the oral cavity explains mucosal melanomas.

3. *Two potential explanations are:*

a) The UV-related oral/nasal mucosal melanomas are metastases from a sun-exposed site such as the lip.

b) Something in the Chinese diet, or smoking or a habit like chewing betel nuts, triggered chemically-induced UV-like DNA photoproducts in these patients, analogous to the later steps of a mechanism recently reported (Science 347:842, 2015). This is more hypothetical, as it would require a previously unknown trigger and it leaves unexplained the oral/nasal melanomas that didn't have UV signatures.

4. *At a minimum, the paper needs to state that only a small fraction of the mucosal melanomas of gingiva, nasal cavity, and oropharynx were sun-related and that this is a bit of a mystery because the xeroderma pigmentosum data suggest that these sites are not directly exposed to sunlight.*

Response: We thank the reviewer for raising this important point and we agree that not all tumours appearing at potentially sun-exposed sites exhibit a dominant SBS7 signature. We agree that multiple alternative/additional processes may play a role in these tumours including metastasis, chemical-induced UVR-like DNA damage or underlying germline status. Importantly, the observation is consistent with our previous study in cutaneous melanoma, where we exposed a BRAF^{V600E}-driven mouse melanoma model to UVR, and observed that a some melanomas did not exhibit typical UVR-associated DNA damage (Trucco et al, 2018, *Nature Medicine*, 2019 Feb;25(2):221-224). We agree with the reviewer that this is a

mystery, in response we have added a comment in the *Discussion* to acknowledge the point (starting page: 9 line 234).

Comment 5: *Whatever the biological explanation, it does remain clinically useful to identify those mucosal melanomas that have sunlight-like mutations, using the small gene panel that the authors describe, because these appear to also have mutated genes for which there are therapeutic approaches.*

Response: We thank the reviewer for their appreciation of the clinical implication of our findings, and the potential utility of the gene panel in differentiating between the UVR and non-UVR melanomas.

Comment 6: *I also have a two suggestions for conveying the message more clearly.*
Title:

The reader will assume that the term "subtype" means a histological subtype, but what is meant is "a subset of mucosal melanomas". For the reasons outlined above, one can't even say "on sun-exposed body sites", although that would be true for conjunctiva and lip.

Response: We thank the reviewer for this suggestion, and in response we have changed the title of the manuscript to: "*Ultraviolet radiation drives DNA damage in a subset of mucosal melanomas*".

Comment 7: *Abstract and Conclusion:*

"... could benefit from treatments currently used for cutaneous melanoma".

My impression is that melanoma treatments are not chosen based on a UV origin of their mutations. The authors need to clarify whether they are thinking that:

a) the same genes are hit as in cutaneous melanoma and so targeted therapies might work in these patients or

b) the high mutation burden suggests that immunotherapies might be effective in these patients.

Response: We thank the reviewer for the comment which, combined with Reviewer #1 comment 2, prompted us to clarify the clinical implications of our study. A detailed *Discussion* is now added in the manuscript (page 8, line 200).

Reviewer #3

Mucosal melanomas are conventionally considered a separate "species" of melanomas that are distinct from the cutaneous melanomas. In this manuscript, the authors have presented whole genome sequencing analysis of conjunctival mucosal melanomas and shown that majority of these melanomas harbor mutational signatures that suggest that these melanomas are driven by ultraviolet radiation (UVR), akin to their cutaneous counterparts, but unlike the mucosal melanomas from sun-protected areas. The authors conclude that UVR induces melanomagenesis independently of the tissue microenvironment. The study design, implementation, data acquisition and analyses, and interpretations of the data are of high quality.

Comment 1:

However, this study is entirely descriptive with a predictable outcome, i.e. that UVR-exposed mucosal melanomas harbor UVR mutational signatures. The paper lacks novelty, innovation, and palpable biological insights.

Response: We thank the referee for the appreciation of our research strategy and analytical rigor. However, we respectfully highlight that although our study does not provide mechanistic insights, as that is beyond the scope of our study, it is the first study showing genomic patterns of UVR-exposed mucosal melanomas that is strikingly similar to the UVR-exposed cutaneous melanomas. Our study therefore delineates UVR-induced disease aetiology in mucosal melanoma that is more akin to cutaneous melanomas. These findings may have important clinical implications for this subset of mucosal melanoma patients, as they may benefit from treatment options, including both targeted and immunotherapy, only currently available to cutaneous melanoma patients. We have elaborated on this more extensively in the *Discussion* (page 8, line 200).

We also posit that this study provides novel insight into the mutational processes underlying both cutaneous and mucosal melanoma, as we observe that the UVR-induced DNA damage can potentially impose on a mucosal melanoma a disease aetiology more akin to that of a cutaneous melanoma.

REVIEWERS' COMMENTS:

Reviewer #2 (Remarks to the Author):

The revision is a significant improvement and with only minor additional changes will satisfy this reviewer's concerns.

1. The authors' clarification of the clinical implications of these findings – that mucosal melanoma patients should not be excluded from certain tests or treatments – is a significant improvement for this reviewer's concerns in that regard.
2. The handling of the UV exposure question is improved, but still a little rocky. This is partly because the wording changes have not been made everywhere, making the logic arrive out of order. Below, I suggest some re-wording to correct this.
3. The now-clearer manuscript reveals that the authors are underemphasizing an important result that perhaps clarifies the UV-exposure and mechanism questions and probably rules out the metastasis option I'd suggested: The UV-related mucosal tumors have both the UV-like point mutations typical of cutaneous melanomas and the large structural mutations typical of mucosal melanomas. (This is never stated so directly. It is treated more as "Unfortunately you can't use structural mutations to distinguish the categories.") That finding suggests that:
 - a) the UV-related tumors in the mucosa are genuine mucosal cells rather than metastases from elsewhere and
 - b) the UV-related mutations could have arisen early but did not make a melanoma, after which the structural mutations (known to be needed for mucosal melanomas) arose and converted one UV-mutant cell to mucosal melanoma.

The implication is then that a mucosal cell in a region receiving sun exposure could accumulate UV-induced mutations and, over time, expand into a large clone of mutant cells extending well into the sun-protected area. This could be due to neutral drift, but the presence of driver mutations suggests an active process. The best examples of this clonal expansion are in Barrett's esophagus (eg Maley et al, Cancer Res 64:3414, 2004) but it is also seen in skin (eg Klein et al, PNAS 107:270, 2010). Afterward, a mucosal cell in a dark region could develop into a mucosal melanoma that is UV-like. This is still a hypothesis and so best in the Discussion, but it seems a better alternative than metastasis or UV reaching the back of the mouth.

Suggestions

1. Title Should be "drives mutations"; nowhere in this paper is DNA damage measured.
2. Abstract suggestion. "could benefit from treatments" might be more impactful if re-stated as "should not be excluded from treatments"
3. line 121 and below. This entire section comes before the section discussing UV-exposed and unexposed body sites, so it should stick with what is known: UV mutation signature status rather than "UVR-exposure". So:

122 are independent of UV signature mutation status
129 between cohorts containing UV signature mutations or not

[Note that the Fig2a being referred to does define the cohorts in terms of the UV signature SBS7v2. Also need to fix the typos here "SBSv7".]
Now the next section on sun exposure follows logically.

158 ... than tumor site and we will now refer to these tumors as UVR-driven mucosal melanomas.

193-195 Here would be a good place for a clear statement along the lines of "... two broad groups, one containing only the large structural mutations characteristic of mucosal melanomas and the other in addition containing UV-related mutations typical of cutaneous melanomas".

213 Suggestion. I suspect the authors intend "as high as half" but to the reader it sounds like "only half".

214 Typo "in", not "in to"

223 "dominant" is confusing and the technical genetics meaning has not been demonstrated. "additional" would be better.

238 This might be the place to delete metastases and insert the evidence for large clonal expansions from regions closer to sun exposure. By the way, the high UV mutation burden would seem to rule out another idea, that mucosal cells in the back of the mouth received a low UV dose that happened to make a rare UV mutation that led to cancer.

We thank the reviewer for their time in reviewing the revised manuscript and their insightful comments, which have helped us to further improve the quality of the manuscript. Below we provide responses to each comment.

Reviewer #2:

General comment. The revision is a significant improvement and with only minor additional changes will satisfy this reviewer's concerns.

Comment 1. The authors' clarification of the clinical implications of these findings – that mucosal melanoma patients should not be excluded from certain tests or treatments – is a significant improvement for this reviewer's concerns in that regard.

Response. *We are pleased that the Reviewer finds the manuscript improved, and the clinical implications clarified.*

Comment 2. The handling of the UV exposure question is improved, but still a little rocky. This is partly because the wording changes have not been made everywhere, making the logic arrive out of order. Below, I suggest some re-wording to correct this.

Response. *We thank the Reviewer for these suggestions and have taken them board, as noted in the responses below.*

Comment 3. The now-clearer manuscript reveals that the authors are underemphasizing an important result that perhaps clarifies the UV-exposure and mechanism questions and probably rules out the metastasis option I'd suggested: The UV-related mucosal tumors have both the UV-like point mutations typical of cutaneous melanomas and the large structural mutations typical of mucosal melanomas. (This is never stated so directly. It is treated more as "Unfortunately you can't use structural mutations to distinguish the categories.") That finding suggests that:

- a) the UV-related tumors in the mucosa are genuine mucosal cells rather than metastases from elsewhere and
- b) the UV-related mutations could have arisen early but did not make a melanoma, after which the structural mutations (known to be needed for mucosal melanomas) arose and converted one UV-mutant cell to mucosal melanoma.

The implication is then that a mucosal cell in a region receiving sun exposure could accumulate UV-induced mutations and, over time, expand into a large clone of mutant cells extending well into the sun-protected area. This could be due to neutral drift, but the presence of driver mutations suggests an active process. The best examples of this clonal expansion are in Barrett's esophagus (eg Maley et al, Cancer Res 64:3414, 2004) but it is also seen in skin (eg Klein et al, PNAS 107:270, 2010).

Afterward, a mucosal cell in a dark region could develop into a mucosal melanoma that is UV-like. This is still a hypothesis and so best in the Discussion, but it seems a better alternative than metastasis or UV reaching the back of the mouth.

Response. We thank the Reviewer for this interpretation of our data, which aligns with our own. In response and as per the Reviewer's suggestion, we have removed reference to in-transit metastases and added the following sentence in the Discussion:

"Whilst imprecision in site reporting may play a role in this, one possibility is that mucosal cells in sun-exposed sites may accumulate UVR-induced mutations and expand into large clones of mutant cells, extending into sun-protected areas where such cells could further develop into a melanoma. Clonal expansions of this nature have previously been reported in the skin ²⁷ and in Barrett's esophagus²⁸."

Suggestions

1. Title Should be "drives mutations"; nowhere in this paper is DNA damage measured.

Response. We have amended the title as suggested.

2. Abstract suggestion. "could benefit from treatments" might be more impactful if re-stated as "should not be excluded from treatments"

Response. We thank the Reviewer for this suggestion and have changed the abstract accordingly.

3. line 121 and below. This entire section comes before the section discussing UV-exposed and unexposed body sites, so it should stick with what is known: UV mutation signature status rather than "UVR-exposure".

So: 122 are independent of UV signature mutation status

Response. We have amended this to read "independent of UVR mutation signature status"

4. 129 between cohorts containing UV signature mutations or not [Note that the Fig2a being referred to does define the cohorts in terms of the UV signature SBS7v2. Also need to fix the typos here "SBSv7".]

Response. We have amended this section to read: "Critically, we observed no significant differences between the SBS7v2-dominant and the non-SBS7v2 cohorts (Fig 2a, Extended Data Fig 3a), and similarly no significant differences in copy number variations (Fig 2b)."

We have also corrected the typos noted by the Reviewer.

5. Now the next section on sun exposure follows logically.

158 ... than tumor site and we will now refer to these tumors as UVR-driven mucosal melanomas.

Response. *We have amended the text to read: "than tumor site and henceforth we refer to these tumors as UVR-exposed mucosal melanomas."*

6. 193-195 Here would be a good place for a clear statement along the lines of "... two broad groups, one containing only the large structural mutations characteristic of mucosal melanomas and the other in addition containing UV-related mutations typical of cutaneous melanomas".

Response. *We have reworded the section as follows:*

"Our results extend previous studies suggesting that conjunctival mucosal melanomas have similar genomes to common cutaneous melanoma^{12,13,14}. We also show that like common cutaneous melanoma³, mucosal melanomas present two broad groups, one with SBS7v2 predominance that appears to be UVR-driven, and one that is not UVR-driven, but curiously, both groups present the large structural alterations more commonly observed in mucosal melanoma."

7. 213 Suggestion. I suspect the authors intend "as high as half" but to the reader it sounds like "only half".

Response. *We have amended this to read: "...in advanced mucosal melanomas are up to half of that seen in non-glabrous skin melanomas"*

8. 214 Typo "in", not "in to"

Response. *This has been corrected.*

9. 223 "dominant" is confusing and the technical genetics meaning has not been demonstrated. "additional" would be better.

Response. *This has been amended as recommended.*

10. 238 This might be the place to delete metastases and insert the evidence for large clonal expansions from regions closer to sun exposure. By the way, the high UV mutation burden would seem to rule out another idea, that mucosal cells in the back of the mouth received a low UV dose that happened to make a rare UV mutation that led to cancer.

Response. *We concur and this fits with our own thoughts. As noted above, we inserted the following lines at this location:*

"Whilst imprecision in site reporting may play a role in this, one possibility is that mucosal cells in sun-exposed sites may accumulate UVR-induced mutations and expand into large clones of mutant cells, extending into sun-protected areas where such cells could further develop into a melanoma."

Clonal expansions of this nature have previously been reported in the skin 27 and in Barrett's esophagus^{28}.”